# Route selection in non-Euclidean virtual environments

**Alexander Muryy, Andrew Glennerster** *

School of Psychology and Clinical Language Sciences, University of Reading, Reading, United Kingdom

* a.glennerster@reading.ac.uk

## Abstract

The way people choose routes through unfamiliar environments provides clues about the underlying representation they use. One way to test the nature of observers' representation is to manipulate the structure of the scene as they move through it and measure which aspects of performance are significantly affected and which are not. We recorded the routes that participants took in virtual mazes to reach previously-viewed targets. The mazes were either physically realizable or impossible (the latter contained 'wormholes' that altered the layout of the scene without any visible change at that moment). We found that participants could usually find the shortest route between remembered objects even in physically impossible environments, despite the gross failures in pointing that an earlier study showed are evident in the physically impossible environment. In the physically impossible conditions, the choice made at a junction was influenced to a greater extent by whether that choice had, in the past, led to the discovery of a target (compared to a shortest-distance prediction). In the physically realizable mazes, on the other hand, junction choices were determined more by the shortest distance to the target. This pattern of results is compatible with the idea of a graph-like representation of space that can include information about previous success or failure for traversing each edge and also information about the distance between nodes. Our results suggest that complexity of the maze may dictate which of these is more important in influencing navigational choices.

## 1 Introduction

In order to navigate successfully in a 3D environment, human participants have to develop a mental representation of the scene, locate themselves in the representation and plan optimal actions to reach a target. The exact form that such a mental spatial representation might take is still debatable. One view is that the spatial representation corresponds to a cognitive map [1–3], i.e. a stable 3D reconstruction of the environment (whether accurate or not). This provides the most complete description of the environment and can be used for versatile spatial tasks such as planning an optimal route, exploring novel shortcuts or pointing to unseen targets. It could be constructed by means of path integration [4] and fully working implementations of this model are now common in the computer vision and robotics literature based on visual SLAM (Simultaneous Localisation and Mapping) [5] which integrates information from views

**Funding:** AG received EPSRC/Dstl funding: Engineering and Physical Sciences Research Council and Ministry of Defence, EP/N019423/1,

https://epsrc.ukri.org/ and https://www.gov.uk/government/organisations/defence-science-and-technology-laboratory. The funders had no role in study design, data collection and analysis, decision to publish, or preparation of the manuscript. AG received EPSRC funding: Engineering and Physical Sciences Research Council, EP/K011766/1, https://epsrc.ukri.org/. The funders had no role in study design, data collection and analysis, decision to publish, or preparation of the manuscript. AG received AHRC funding: Arts and Humanities Research Council, AH/N006011/1, https://ahrc.ukri.org/. The funders had no role in study design, data collection, decision to publish, or preparation of the manuscript.

**Competing interests:** The authors have declared that no competing interests exist.

over multiple vantage points. It has been argued that in small and relatively simple environments such as 'vista spaces' participants have access to a relatively accurate cognitive map within a confined region [6, 7] although even in the case of vista spaces there is dispute about whether the underlying representation in this case is Euclidean [8], i.e. corresponds to a rigid 3D reconstruction. However, in larger and more complex environments there is greater agreement that Euclidean reconstruction is a poor model. For instance, the perceived length of a route depends on the number of turns and decision points it contains [9–11], angular and directional judgments are highly inaccurate [7, 12–14] and perceived angles between junctions are biased towards 90° [10, 15]. Hence, while mental representations of small open environments can often appear to be consistent locally, participants typically have difficulties integrating local representations into a single global representation (as has been argued for other primates, too). In particular, performance in large environments is much more likely to be compatible with a distorted or globally inconsistent map [15, 16]. This led Kuipers [17] to suggest that the concept of a global 'Map in the Head' should be replaced by an 'Atlas in the Head', with many local maps on separate sheets. Similar ideas of independent reference frames consisting of multiple vista spaces were also proposed in more recent studies [6, 7]. It is not clear how these local representations are used by participants when they are confronted by a spatial task (such as pointing) that forces them to integrate information across different local reference frames except that, as Meilinger and colleagues say [6], pointing appears effortful and performance depends on many factors such as the order in which the route was learned. Experimental evidence suggests that performance in this case relies on a representation (or a process of accessing information from a representation) that is not only distorted but also inconsistent with the idea of a single global map [18–21].

In an early seminal paper, Siegel and White [22] suggested that, in large-scale environments, spatial representation develops gradually and goes through three main phases: landmark knowledge (salient features), route knowledge ('place-goal-action associations') and survey knowledge (construction of a cognitive map) [23]. Developing this type of idea, Kuipers [17] suggested that, as more information becomes available about an environment, 'topological connections can be strengthened into relative-position vectors' and then, ultimately, a representation uniting multiple frames of reference. He emphasized the co-existence of multiple strategies based on different levels of detail which he described as a cognitive map having 'many states of partial knowledge'. Montello [24] criticized Siegel and White's idea, pointing out that there can be gradual 'quantitative accumulation and refinement of metric knowledge'. Ishikawa and Montello [14] set out to test the developmental progression of representations that Siegel and White and others have advocated and found very little learning across trials (although no feedback was given). They emphasised the fact that some individuals acquired 'surprisingly accurate metric knowledge, even relatively quickly' relating locations between which they had not travelled directly. In line with this finding, when Weisberg and colleagues [25–27] tested a large number of participants in virtual reality (VR), they found that there was significant variation in the ability of people to integrate spatial information across routes: participants' pointing performance within a familiar route was not necessarily a good predictor of their ability to point between targets on two different familiar routes.

Warren [29] has drawn together much of the literature on navigation in Euclidean (physically possible) and non-Euclidean environments arguing that the evidence points to humans using a 'labelled graph' (Chrastil and Warren [28], Strickrodt et al [20], Warren et al [19]). This lies between a topological graph and survey knowledge because each edge of the graph can include information about the length of the path connecting those two nodes and, as someone becomes more familiar with an environment, there can be information stored about the angle between edges. Warren [29] emphasizes that a labelled graph can become more and

more accurate with experience: "*One would expect edge weights and node labels to become more accurate and precise with repeated exposure to an environment,*" (p4). In theory, the information about edges can become so accurate that tasks such as pointing from the current node to an object at another node can be as accurate as it would be based on a Euclidean map, making it impossible to distinguish between a graph and a map for such tasks. A very similar spectrum has been proposed for the processing of disparity information to guide judgements of ordinal depth, bas relief depth or Euclidean shape [30, 31].

There have been many studies that have explored the extent to which participants can encode actions that have led to a successful result in the past and incorporate this in their representation [32–35]. Marchette et al [34] showed that in a navigational experiment when searching for targets some participants found novel shortcuts easily, while other participants preferred less efficient, but more familiar routes that they had experienced during the learning phase. fMRI analysis showed that participants who preferred shortcuts had a stronger activation in the hippocampal area, while participants who followed the more familiar route had a stronger activation in the caudate which encodes reward. Chrastil and Warren [28] review a hierarchy of tasks and corresponding representations that would support such tasks, where route knowledge (in our case, knowing whether to go left or right at a junction to get to a goal) is lower in the hierarchy than knowing a topological map of a maze which would allow observers to take topological shortcuts (i.e. routes traversing a smaller number of edges). Accurate pointing and reliable identification of novel shortcuts are higher in the hierarchy than route knowledge, as both require the observer to do more than simply follow previously rewarded routes. Interestingly, in the reinforcement learning literature there has been a recent focus on representations that are similar to the 'response-like' model in that they learn what action to carry out at each decision point (given a particular goal) rather than computing a global map [36].

In this paper, we build on our previous study of human pointing errors in a virtual maze [18] which, like the current study, examined the consequences of exploring a physically impossible maze. The maze had long corridors with many turns in a way that could not be realized in the real world ('wormholes'), similar to the manipulations many other researchers have used to explore spatial behaviours in non-Euclidean environments [8, 19, 37, 38]. The conclusion of our previous paper was that the most likely explanation of the data in this type of condition was that participants relied on a representation that has no Euclidean interpretation. The current paper examines the performance of the same participants in the same experiment but instead of analysing the pointing responses we report the ability of participants to find the shortest distance through a maze to a target. This task is suited to finding out what information participants use to choose a path when they are at a junction, not to finding out whether they use a Euclidean reconstruction or a graph-like representation. Indeed, if observers have a Euclidean representation that includes the target and their current location, and the task is to choose the shortest route using their representation, then they should do that independent of any past experience of reward. A graph-based representation is more flexible. Initially, observers may only store information about whether or not they have travelled down a particular path and whether this led to the object that is their current goal (similar to 'response-learning', [34, 39, 40]). Later, they may add information about the distance between nodes. In the current experiment (to anticipate our results), we find that the more complex the maze, i.e. with wormholes, the more likely participants are to choose previously rewarded routes. In the Discussion, we consider how this relates to the idea that people may begin with a topological graph of connectivity and gradually add information about reward and distance along corridors (edges in the graph) once they gain more experience of the environment.

## 2 Material and methods

### 2.1 Participants

The 14 participants (5 male and 9 female) who completed the experiment were students or members of the School of Psychology and Clinical Language Sciences. All participants had normal or corrected to normal vision (6/6 Snellen acuity or better), one participant wore glasses during the experiment, and all had good stereo-acuity (TNO stereo test, 60 arcsec or better). All participants were naïve to the purpose of the study. Participants were given a one-hour practice session in VR to familiarize them with our set-up using physically possible mazes. We called physically possible mazes 'Fixed', for short, as they did not change as the participant moved around them. 10 potential participants (1 male, 9 female) either experienced motion sickness during the practice session or could not move confidently in VR and thus preferred not to continue at this stage (any participants who were excluded did so before data was collected for either 'base layout' used in the experiment). The higher-than-normal dropout rate is likely to be due to the overall scaling of the scene which results in a conflict between eyeheight and other cues to scale (discussed in Section 3). Altogether, there were 7 sessions (including the practice), each of about 1 hour, conducted on different days. Participants were advised not to stay in VR longer than 10 minutes between breaks. They received a reward of 12 pounds per hour. All participants gave their written informed consent to take part after reading the Participant Information Sheet. The study received approval of the Research Ethics Committee of the University of Reading.

### 2.2 Experimental set-up

The Virtual Reality laboratory was equipped with a Vicon tracking system with 12 infrared cameras (T20 and Bonitas). We used an nVision SX111 head mounted display with a large field of view (111˚ horizontally with a binocular overlap of 50˚). The resolution of the LCD displays was 1280 by 1024 pixels. The headset was calibrated using the method described in [41] in order to minimize optical distortions in the stimuli. We have measured the motion-to-photon latency of our VR system with the nVis SX111 display as 40ms [42]. The HMD was connected via a 4m-long video cable to a video controller unit on the ceiling. The Vicon tracking system (Tracker 3.1) provided an estimate of the position and orientation of the headset with a nominal accuracy of ±0.1 mm and 0.15˚ respectively at a frequency of 240Hz and relayed this information to a graphics PC with a GTX 1080 video card. The stimuli were designed in Unity 3D software [43] and rendered online at 60fps. Participants were allowed to walk freely and explore the virtual environment in a natural way, although they had to hold the HMD video cable behind them and had to take care that the cable did not become tangled as they walked. The experimenter was always close by to ensure that the cable remained behind them. The physical size of the labyrinth was limited to a 3 by 3m region in the lab. The virtual labyrinth was originally a 5 by 5m environment with corridors in the maze 1m wide. In order to fit in the 3 by 3m space, the labyrinth was shrunk to 0.6 scale (e.g. 60cm wide corridors) which meant that the floor was displayed about 1m below eye height. Participants generally found this acceptable and did not notice that the room was not normal size, consistent with previous reports [8]. During the experiment, participants wore a virtual wristband that provided information about the task (shown, for illustrative purposes only, in the bottom-right corner of Fig 1B). In the pointing phase of the experiment, participants used a hand-held 3D tracked pointing device to point at targets. In VR, the pointing device was rendered as a small sphere (R = 5cm) with an infinitely long ray emanating from it in both directions, although the ray could not be seen beyond the corridor walls. Text was displayed on a panel attached to the ray providing instructions (e.g. 'point to Red'). The 6-degrees-of-freedom pose of the cyclopean

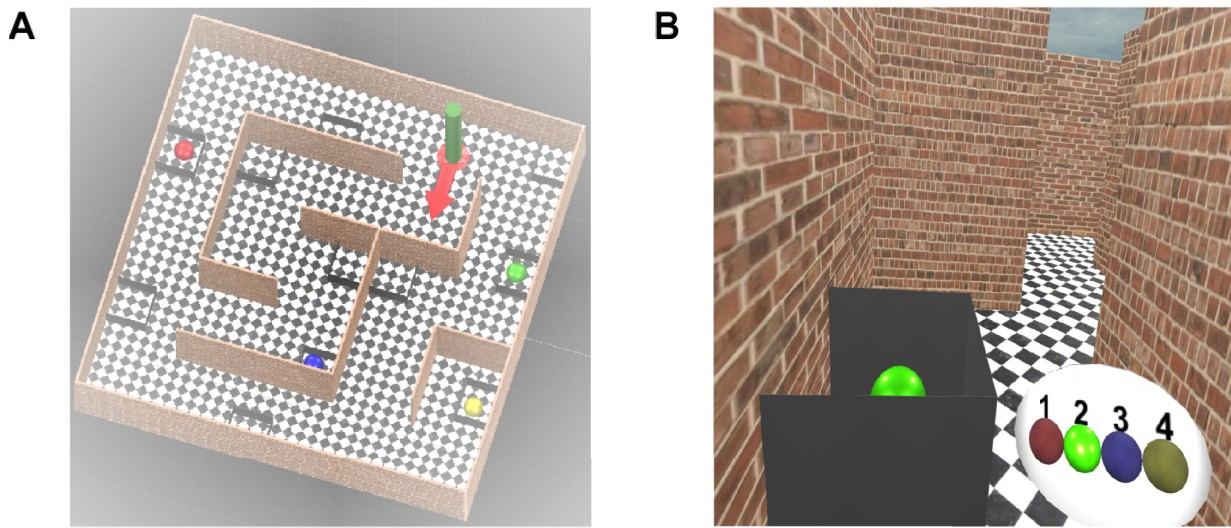

**Fig 1. Views of the labyrinth.** A) View from above. B) First person view. The green target is visible inside a grey box. The target sequence is shown on the wrist-band in the bottom-right corner and the current target is highlighted (Green). A movie version is included in the (S1 Video) for the Fixed condition.

point (a point midway between the eyes), together with the orientation of the headset was recorded on every frame (60 fps).

## 2.3 Stimuli

We designed two general layouts of the virtual labyrinth (Layout 1, shown in Figs 1 and 2, Layout 2 shown in the S1 Fig). Each of these general layouts could be modified by the addition of wormholes. The virtual environment could be subdivided into 25 (5x5) elementary squares each having a size equal to the corridor's width. Initially, the environment consisted only of a chequered floor and a green cylinder, indicating the start location. The participant walked into the green cylinder, faced in the direction of the red arrow (Fig 1A) and then the green cylinder and red arrow disappeared, so that the starting location was not marked during the exploration

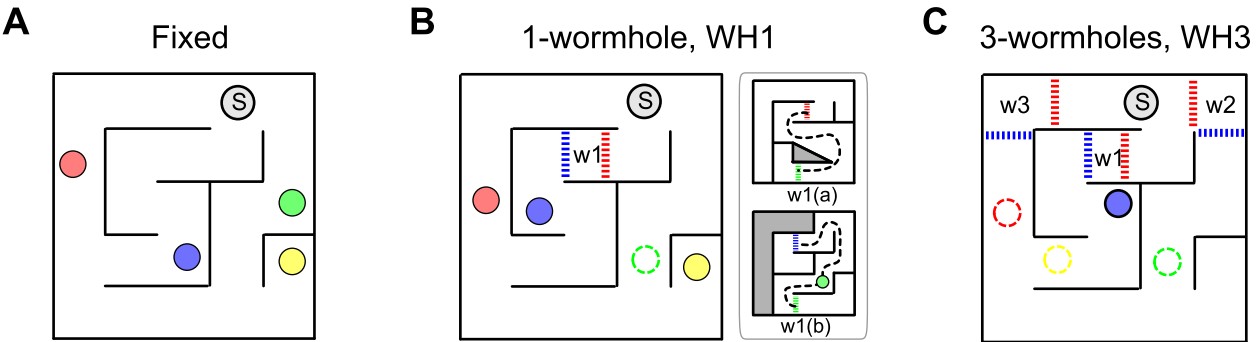

**Fig 2. 'Fixed' and wormhole conditions.** The general layout (containing Start, which is marked as 'S') remained constant between conditions. A) 'Fixed' condition in Layout 1. B) One-wormhole condition in Layout 1; the green target is inside the wormhole and the red and blue dashed lines show the location of the triggers to change the virtual environment to one of the scenes indicated in the subplot w1(a) or w1(b). The only region in which a participant could walk once they entered a wormhole is shown by the black dashed line. See text for details. C) Three-wormhole condition for Layout 1; red, green and yellow targets are inside wormholes (for details of the three wormholes see S1 Fig). For movies illustrating particpant trajectories in the Fixed, one-wormhole and three-wormhole conditions and for the layout of the maze in Layout 2, see (S2 Video for the Fixed scene shown in A, S3 Video for the one-wormhole condition shown in B and S4 Video for the 3-wormhole condition shown in C).

phase. The labyrinth contained 4 target objects (red, green, blue and yellow spheres) hidden inside open grey boxes, so that they could be seen only from a short distance (Fig 1B). Other empty grey boxes were added as distractors.

For each labyrinth, we were able to increase the complexity of the environment by extending the length of the corridors with non-metric 'wormholes', see Fig 2B and 2C (for details of three-wormhole condition and Layout 2 see Supplementary Material). There were three conditions per Layout: one 'Fixed' (i.e. rigid and unchanging as the participant explored the maze), one containing one wormhole and another containing three wormholes. Colored circles in Fig 2 show the location of the targets and 'S' shows the Start. In the wormhole conditions, the dashed lines acted as invisible triggers: when a participant crossed this line, the environment changed as shown in the sub-plots although the changed regions were always out of sight at the moment the participant passed through the trigger so there was no visible indication that anything had changed. For instance, in the one-wormhole condition shown in Fig 2B, if a participant were to cross the trigger indicated by the red dashed line, the environment would change to schematic W1(a); if the participant continued walking down the path through the wormhole (e.g. along the dashed black line) and crossed the green trigger line, the environment would change again to schematic W1(b), then if the participant crossed the blue trigger line he or she would exit the wormhole and the environment would change back to the original layout. Note that the same is true if the participant were to enter the wormhole the other way: they would then move from W1b scene to W1a scene and back to the original base layout.

For both Layout 1 and Layout 2, the wormhole conditions were derived from the layout of the Fixed condition, as shown in Fig 2. One way to think of the wormholes is as generating a new floor in a building and suddenly transporting the participant to a new floor. According to this analogy, for a given Layout (say, Layout 1) the 'ground floor', or base-level layout, of the environment was the same for Fixed, one-wormhole and three-wormhole conditions. The corridors through the wormholes did not have any junctions which meant that the topological connectivity of space was the same in all 3 conditions (although the different coloured targets could be placed at different locations within the maze). The main difference between Fixed and wormhole conditions was the length and configuration of the corridors. The wormholes extended the corridors in a way that made a correct Euclidean representation impossible. For instance, the path through the wormhole in Fig 2B has the shape of a figure of eight, i.e. it crosses itself, although there are no visible junctions along that path, which is physically impossible.

## 2.4 Procedure

Participants followed the instructions they were given, finding the four targets shown on their wristband in the specified order. When they reached the fourth target, they pointed at the other targets and at the Start location but the results of this pointing task are reported in a separate paper [18] so ithey are not described further here, although see Fig 13 for a comparison of the pointing data and the navigational choice data. In the course of one experimental session, which took about 1 hour, participants were tested sequentially on the three types of maze, i.e. Fixed, one-wormhole and three-wormhole conditions, all with the same general layout (i.e. all Layout 1 or Layout 2). This was designed deliberately to help participants to navigate in the more complex environments. The tasks and instructions were identical for all three conditions. The instructions given to participants were to collect all four target objects in a specified order in the most efficient way. 'Collect' meant approach sufficiently close to the target (within a radius of 0.5m from the cyclopean point and within the field of view) which caused its colour to change from bright to dull and, at the same time, the colour of that ball changed in the same way on the wrist-mounted panel. The meaning of 'efficient' was not defined precisely for participants although it was

emphasized to them that they should not hurry and that their performance was not being judged by their speed. 'Efficient' could mean choosing the shortest path, or the smallest number of turns or junctions (i.e. navigational decisions)–this was left to participants to decide.

The first five rounds were a 'learning' phase in which participants always began at the Start location and 'collected' targets in the same sequence Start-Red-Green-Blue-Yellow (S-R-G-B-Y). The purpose of the learning phase was to allow participants to build up a spatial representation of the labyrinth gradually through multiple repetitions of the same navigational task. During the test phase (the last 3 rounds out of a total of 8 rounds), the navigational sequences were changed to three new sequences: Y-G-B-Y-R, R-B-R-Y-G and G-Y-G-R-B. Participants did not have to go to the Start locations at the beginning of a round but instead started at the location where the previous round ended.

Excluding the practice session, each participant carried out 6 experimental sessions, each on a different day. We tested one Layout per session (Layout 1 or Layout 2), testing 'Fixed', 'one-wormhole' then 'three-wormhole' conditions in the session. On different days (sessions) participant was tested on alternating Layouts (Layout 1 then Layout 2 etc). Then participants repeated the sequence for two repetitions, hence 6 days (sessions). S2 Fig lists all 18 conditions that participants experienced (2 Layouts, 3 room conditions ('Fixed', one-wormhole and three wormhole) and three repetitions). The repetitions were not identical because the colours of the targets were switched around, so that on repetition 2 the blue sphere might appear in the box where the red sphere had appeared in repetition 1. Importantly, the structure of the maze and the location of the grey boxes remained the same. This meant that while the instructions remained the same (e.g., in the learning phase, collect targets in sequence R-G-B-Y) the actual routes in the maze to complete those tasks were different on different repetitions. For all subsequent description and figures in the paper, however, in order to make it easier to follow, the colours of the target at each location in the maze or graph remain the same per Layout, independent of the repetition. We also used these labels for the nodes in the analysis.

For the purposes of analysis, we divided participants' movements into discrete steps, as follows. During the experiments, we recorded participants' locations and orientations at 60 frames per second and then converted these trajectory data into topological steps through the maze. For instance, participant P5 made the following steps in Layout 1, Fixed condition (start locations and goal locations are shown in bold):

Learning round 1, task Start-R-G-B-Y: **S** B N1 N2 Y N2 N1 **R** S **G** N2 Y N2 N1 **B** N1 N2 **Y**

Learning round 2, task Start-R-G-B-Y: **S** B N1 **R** N1 N2 **G** S **B** N1 R S G N2 **Y**

Learning round 3, task Start-R-G-B-Y: **S R** S **G** S **B** N1 N2 **Y**

Learning round 4, task Start-R-G-B-Y: **S R** S **G** S **B** N1 N2 **Y**

Learning round 5, task Start-R-G-B-Y: **S R** S **G** S **B** N1 N2 **Y**

Test round 1, task Yellow-G-B-Y-R: **Y** N2 **G** S **B** N1 N2 **Y** N2 G S **R**

Test round 2, task Red-B-R-Y-G: **R** N1 **B** N1 **R** S G N2 **Y** N2 **G**

Test round 3, task Green-Y-G-R-B: **G** N2 **Y** N2 **G** S **R** S **B**

where S is Start and N1 and N2 are the 3-way junctions shown in Fig 3. This labelling of the routes that participants made was a prerequisite to modelling their navigational decisions, as described in the next section.

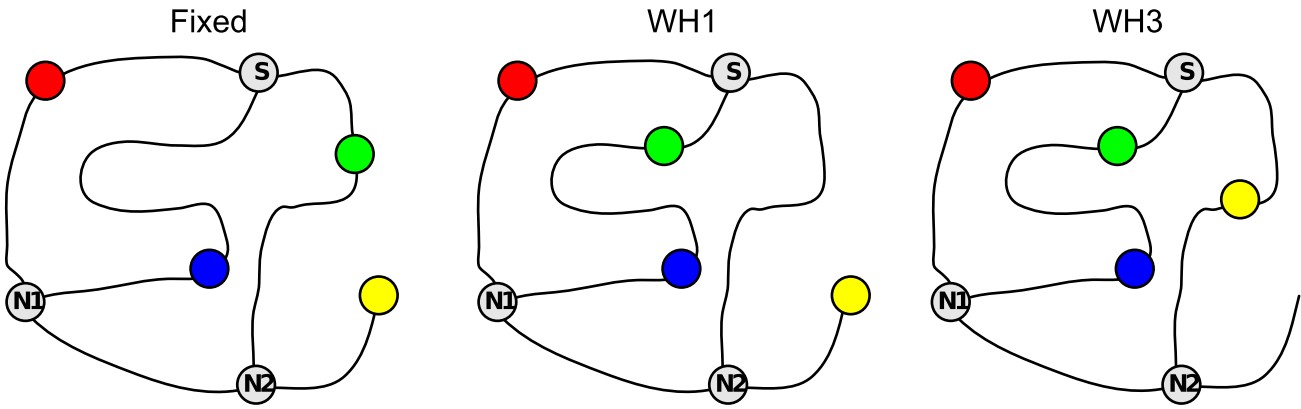

**Fig 3. Topological graphs corresponding to the schematics shown in Fig 2A, 2B and 2C (Layout 1).** Coloured circles represent targets; S, N1 and N2 are 3-way junctions; S is the start location.

## 3 Results and modelling

When participants are allowed to move freely through a maze, it can be challenging to aggregate their data in meaningful ways. Our principal solution to this problem was to compare the likelihood of their navigational decisions under rival models. Before presenting the results of this modelling, there are some general observations that can be made. First, participants' trajectories demonstrate learning, in the sense that trajectories became progressively closer to the shortest metric route during learning. Fig 4 illustrates this pattern for Layout 1. It also shows the increasing length that was required for participants to reach the targets, even by the shortest possible routes, as they go from 'Fixed' to one-wormhole to three-wormhole conditions. Fig 5 illustrates this for a particular task (going from G to Y in this case). It shows how the paths between targets become increasingly convoluted in the wormhole environments even when, from a topological perspective, the task is similar. Fig 6 includes some of the sketches that participants made of the environment, illustrating the confusion that becomes apparent when they have to pinpoint the location of the target spheres on a map (see the coiled lines corresponding to wormhole corridors in Fig 6B and 6C).

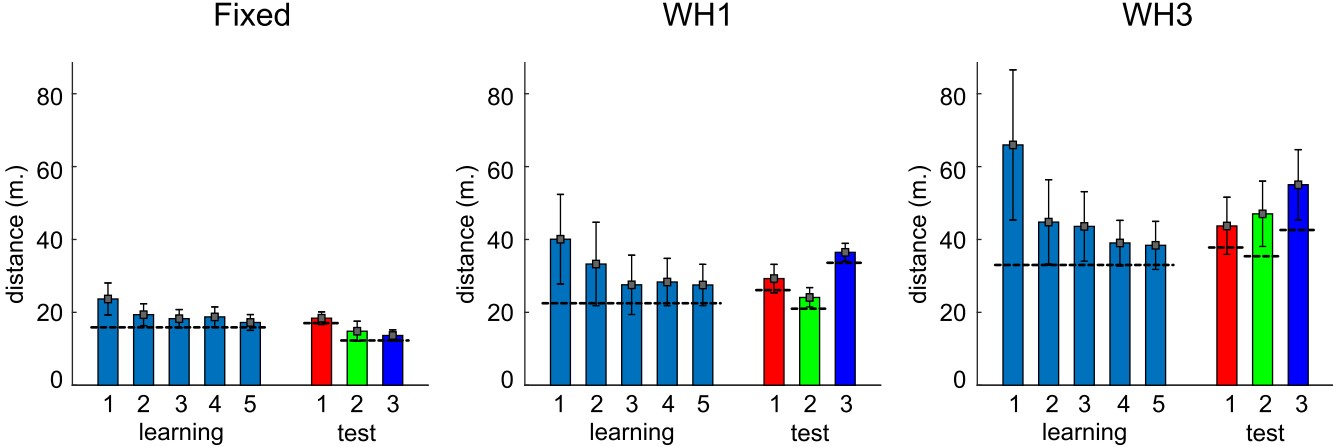

**Fig 4. Travelled distance per round.** Bars show mean distances travelled by all participants (n = 14) in Layout 1, repetition 1. Error bars indicate standard deviations. Horizontal black lines indicate lengths of the shortest path to the target, measured along the middle of the corridors. During the 5 rounds of the learning phase, the task was always the same. During the test phase, participants' tasks were different on every round. The three panels show data from the Fixed, one-wormhole and three-wormhole conditions. Similar plots for all Layouts and all repetitions are shown in S2 Fig.

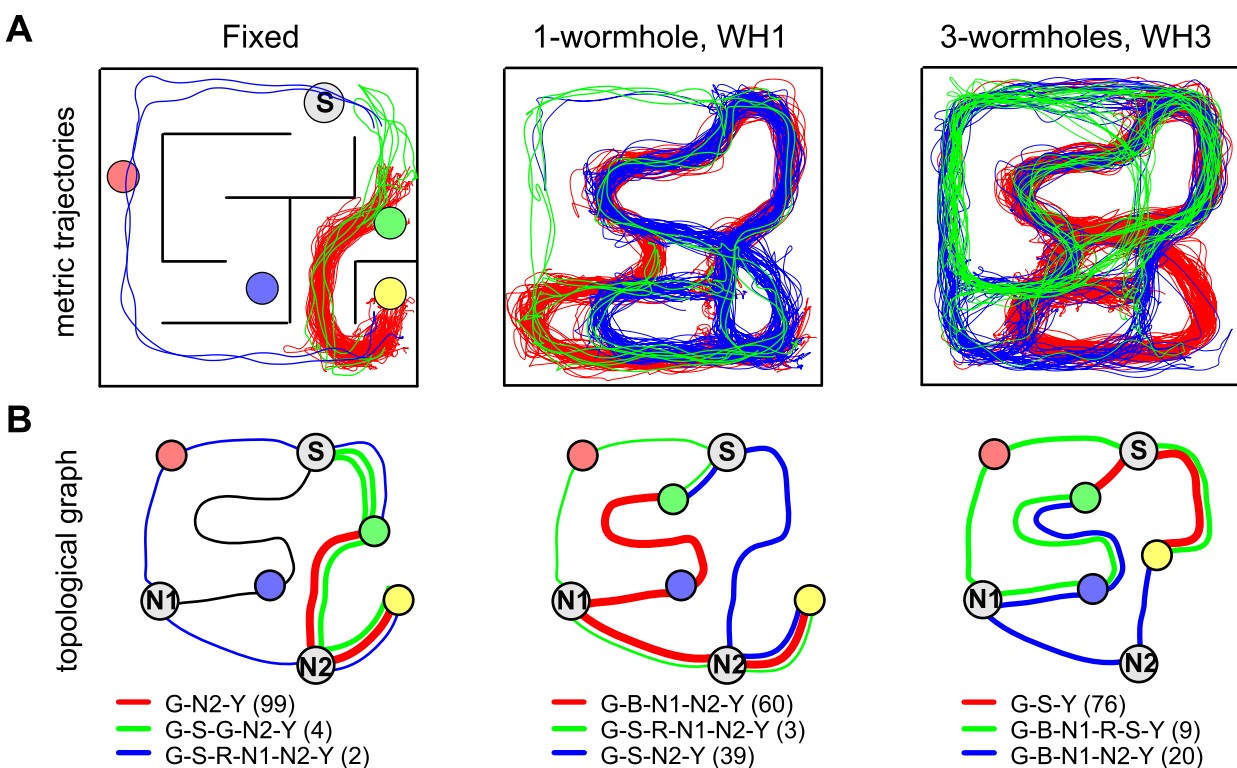

**Fig 5. Examples of a participants' paths.** A) Trajectories of paths taken during one subset of the task, "go from G to Y" in Layout 1 for the Fixed, one-wormhole and three-wormhole conditions. The shortest path is marked in red, while green and blue lines represent alternative routes. Trajectories are drawn in the coordinate frame of the lab. B) Same data shown as a topological graph. Numbers in brackets indicate the number of times each route was taken (all participants, all runs). See Fig 2 for details of the layout in the wormhole conditions.

In the following section, we consider two models. One takes into account the participant's previous experience and whether one path or another was successful in the sense that it led, ultimately, to the goal that the participant had at the time. If so, this model predicts that the path is more likely to be taken during the test phase. We call this a 'Rewarded-choice model'. This approach is somewhat similar to the 'Dual Solution Paradigm' proposed by Marchette et al [34]. Even though in our experiment participants were not restricted in their paths during the learning phase, as they were in Marchette's experiment, it is still possible for us to evaluate the degree of familiarity of the routes that participants took in the test phase. The second model assumes that the participant knows the length of all paths to the goal. We call this the 'shortest distance model'.

## 3.1 Rewarded-choice model

The rewarded-choice model takes into account all navigational decisions that participants took during the learning phase, and the success or otherwise of the choice that they took at any particular junction. It uses this information to predict how they might behave during the three test rounds for that condition. Consider the connectivity matrix for Layout 1 in the one-wormhole condition shown in Fig 7B. This shows which paths are possible between any two nodes in the graph (Fig 7A). The rows represent 'beginning' nodes, i.e. places where the participant has a choice about which way to go. The columns represent 'end' nodes, i.e. where the participant arrives after having made that decision, and a '1' means it is possible to get directly between these two (i.e. there is an edge in the graph between these two nodes). For instance,

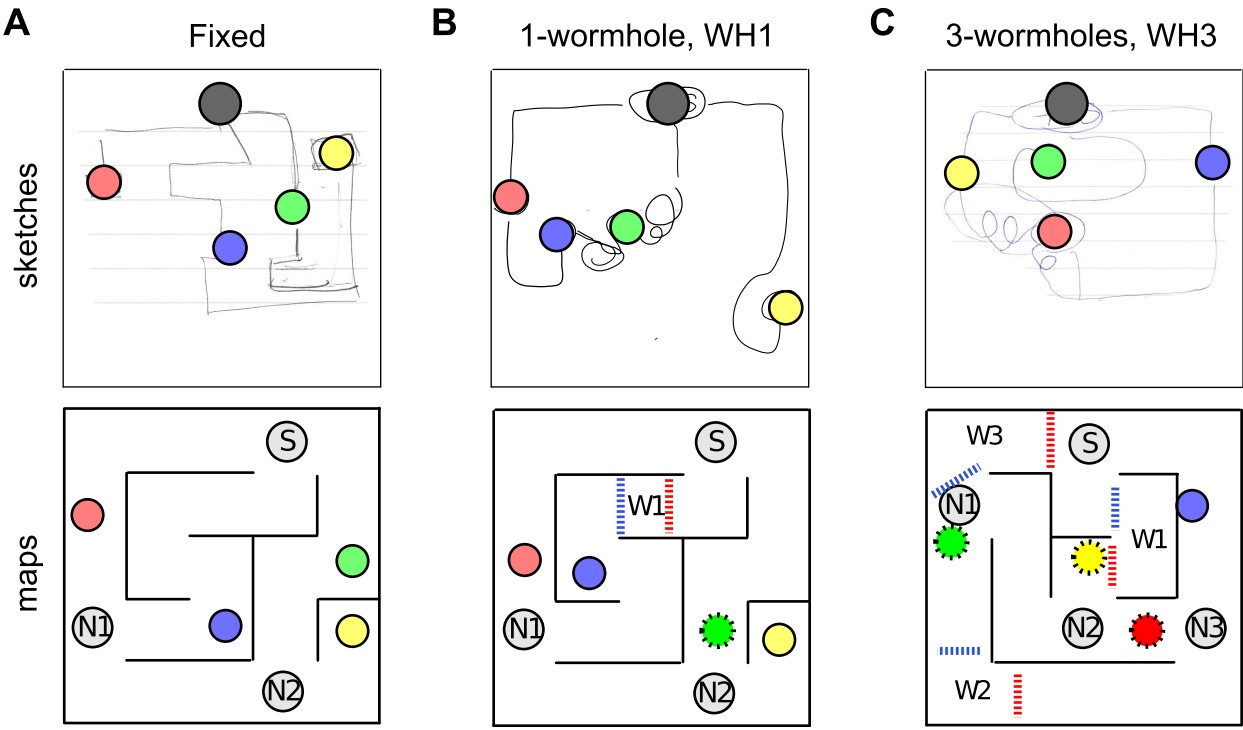

**Fig 6. Sketches drawn by participants.** The black circle indicates the Start location, coloured circles are the targets (added to the sketches for clarity). A) Fixed condition, Layout 1. Notice that the schematic is very accurate except for scale (e.g. length of the corridor with the Yellow target). B) one-wormhole condition, Layout 1. The Green target was inside a wormhole and, from the squiggles connecting it to other targets, the participant appears to be confused about its location on the map and the shape of the corresponding corridor, while Red, Blue and Yellow targets are sketched correctly. C) Three-wormhole condition, Layout 2. The participant makes large errors in the locations of several targets but demonstrates knowledge of topological properties (connectivity between nodes) of the maze. A and B are from Layout 1, C is from Layout 2. More sketches are included S3 Fig.

from the Start node (first row), possible steps are to Red, Green and N2 (columns 2, 3 and 7). If we assume that at the beginning of the learning phase the participant does not have any prior knowledge about the structure of the maze, and thus all decisions about the route are

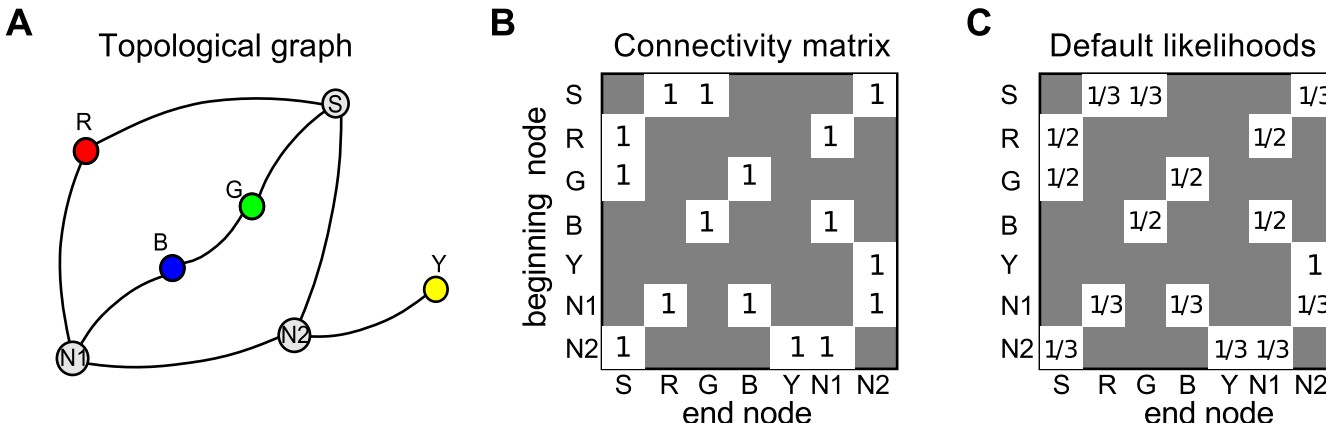

**Fig 7. Connectivity and default decision matrix.** A) Topological graph for Layout 1. 'S' indicates the Start location where participants entered the maze and 'N1' and 'N2' are nodes in the graph indicating 3-way junctions in the maze. B) Corresponding connectivity matrix. C) Default likelihoods of steps, prior to the learning phase.

equally probable, the connectivity matrix of Fig 7B can be converted to a matrix showing the likelihood of taking each path at any given junction as shown in Fig 7C. The probabilities on each row must sum to 1 so, at this default stage, 2-way junctions have a 50% probability for each path and 3-way junctions 33%.

In order to predict the choices that participants will make in the test phase, separate decision matrices are required per participant and per goal (because a participant might be expected to make a different choice at a given junction depending on what their goal was: R, G, B or Y). These were generated as follows. Starting with the default likelihood matrix (Fig 7C, i.e. random choices), the likelihoods associated with each choice were updated in a way that reflected the participant's success whenever they found the target. We re-played all the participants' trajectories during the learning phase. If the participant found the target at the end of a particular route then the next time the participant reached the same junction and had the same goal, the model assumed the participant was more likely to make the same choice again. To explain how this is done in detail, consider an example in which the participant's path goal was R and their path was **Start**-G-B-N1-**R**. Since the Red target was found successfully, the decision matrix is updated by increasing the likelihood of all the decisions that made up that path according to the formula below. The update rule has one free parameter, $\alpha$, that determines the learning rate. Specifically, the likelihood of the steps S-to-G, G-to-B, B-to-N1 and N1-to-R (i.e. steps that successfully led to the goal R) are all increased using the following updating rule:

$p_{i,j} = \frac{p_{i,j}+\alpha}{\sum_{k=1}^{n} p_{i,k}+\alpha}$. This rule updates the likelihood, $p_{i,j}$, of making a step from node $i$ to node $j$, where $\alpha$ is the learning coefficient and $n$ is the total number of nodes (where n = 3 at the Start, N1 or N2 otherwise n = 2). All other elements of row $i$ ($p_{i,m}$, $m \neq j$) should also be updated as $p_{i,m} = \frac{p_{i,m}}{\sum_{k=1}^{n} p_{i,k}+\alpha}$ which ensures that elements in the row sum up to 1 (see Fig 8). This updating is repeated until all the participant's trajectories for the learning phase have been used.

There are choices to be made in deciding how one should build a learning model of this sort. In our implementation, we assumed that participants would notice when they encountered a target *en route* to their specified goal. This means that we update more than one learning matrix simultaneously. So, for example, in the above case of a participant going from Start

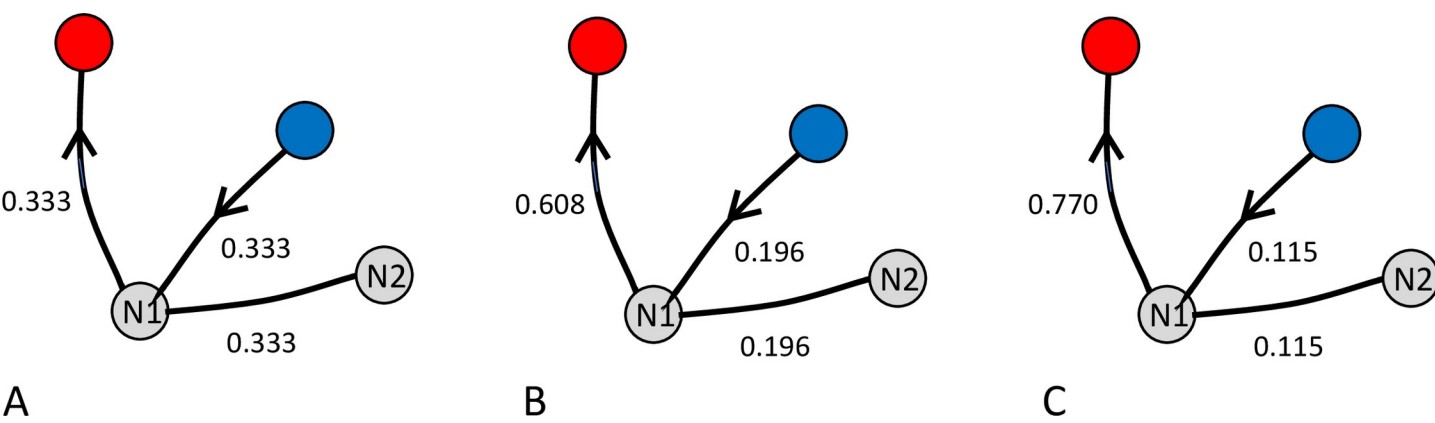

**Fig 8. Illustration of update rule in the rewarded-choice model.** A) At node N1, the default likelihoods for the three potential choices are 1/3 each. Each time the participant chooses the (successful) path from N1 to R (arrows) the likelihoods for the chosen route is increased in the model according to the rule $p_{i,j} = \frac{p_{i,j}+\alpha}{\sum_{k=1}^{n} p_{i,k}+\alpha}$ (see text), where the learning coefficient, $\alpha$, is 0.7 in these examples. The other two routes are updated according to the rule $p_{i,m} = \frac{p_{i,m}}{\sum_{k=1}^{n} p_{i,k}+\alpha}$ (see text). This gives the likelihoods shown in B. When the participant chooses the route N1 to R again, the same rules give rise to the likelihoods shown in C.

to Red by the route **Start**-G-B-N1-**R,** the steps Start-G and G-B are both steps on the way to Blue (so we should update the Blue goal learning matrix) and on the way to Red (so we should also update the Red goal learning matrix). Likewise, we reward the step Start-G in the learning matrix that determines the paths to the Green target. When the participant travelled **Start**-G-B-N1-**R**, we made the choice that, in our model, the reverse route **R**-N1-B-G-Start should be rewarded according to the same rules (i.e. we assumed that people noticed the route that would take them back from Red to Start). The likelihood matrices were filled in by using data from the learning phase only. Note that our model differs from the 'response' model of Marchette et al [34] because in their case the participant had no choice about the route taken during the training phase which meant that, in the test phase, the rewarded route was inevitably the same as the previously-chosen route. That is not the case in our experiment or model.

Fig 9B shows an example of the likelihood matrices calculated for one participant using all their data in the learning phase in the one-wormhole condition (Layout 1). Fig 9A shows an example of a route that that participant took in the corresponding test phase. The task here was to go from Green to Red (notice that this task does not happen during the learning phase). There are 4 possible solutions to this task without loops: 1) G-B-N1-R, 2) G-S-R, 3) G-B-N1-N2-S-R, and 4) G-S-N2-N1-R. In this example, the participant chose the first path, shown in Fig 9A and by the red outlines in Fig 9B. Notice that steps along this path have the highest likelihood in the corresponding matrix (Fig 9B), which illustrates that, in this example, the participant's behaviour during the test phase is consistent with their experience during the learning phase.

## 3.2 Shortest-distance model

The other model is much simpler to describe. The likelihood of a decision under the shortest-distance model can be calculated in the following way. For each binary decision point (i.e. 3-way junction) we found all paths to the goal via the left and right path from the current node (backward steps were not allowed). Then, we found the shortest metric path for each of the

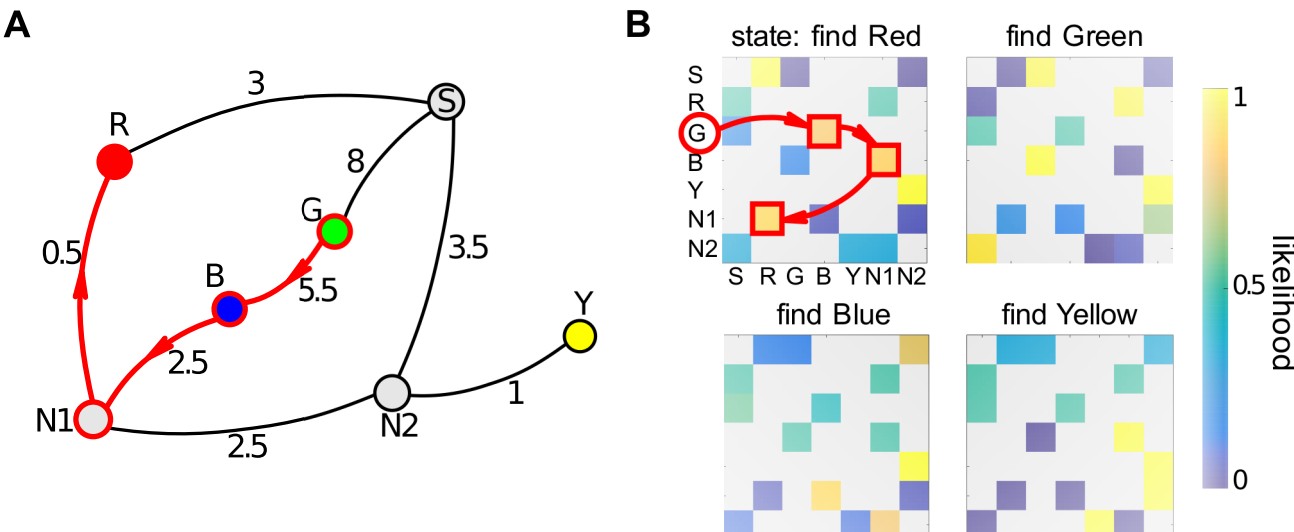

**Fig 9. Constructing a learning matrix.** A) A topological graph for Layout 1 including, along each edge, the distance (in metres) between nodes. Red arrows show an example of the participant's task during the test phase: 'go from Green to Red'. B) Likelihood matrices per target for one participant after they had completed the learning phase. These matrices show the likelihoods according to the rewarded-route model as described in the text. The highlighted elements of the matrix show the likelihoods of the steps shown in A.

two (via left and via right) and calculated their lengths $D_L$ and $D_R$. One option is to assign a probability of 1 in the model to the shortest of these choices (eg the left path) and a probability of zero to the other choice but our model assumed that there was noise on the estimate of lengths $D_L$ and $D_R$ so the probabilities were non-binary. Specifically, we assumed that estimates of the path length are subject to Gaussian noise whose standard deviation is proportional to overall route length (Weber's law): $\sigma_L = \beta^* D_L$, $\sigma_R = \beta^* D_R$, where $\beta < 1$ is a free parameter. The likelihood of taking the shortest route can then be estimated according to the overlap of the two distance estimate distributions. This is $p_{shortest} = 1 - S_{intersection}/2$, where $S_{intersection}$ is the area of the intersection of the two Gaussians and, since there are only two options, $p_{longest} = 1 - p_{shortest}$.

It is important to note that the perceived size of the maze for all participants was determined by eyeheight cues (i.e. participants assumed that their feet were at the level of the floor and that the rest of the scene was scaled accordingly). In fact, as described in the Methods, the virtual floor was 0.6 times the true distance below the eye and the whole scene 0.6 times the normal size so there is a conflict between idiothetic cues from proprioception (distance walked) and interocular separation (baseline cues, as discussed in the Introduction) about the size of the scene and, conversely, these two competing scales give conflicting information about the distance in metres that the participant has walked (see Svarverud et al [8] for discussion of combination of these cues). However, any effects of such conflict would be expected to be the same in the 'Fixed', one-wormhole and three-wormhole conditions.

## 3.3 Model comparison

We compare the performance of the two models in predicting the binary choices participants made during the test phase (the last 3 rounds of 8), i.e. at each 3-way junction (we assumed that they did not go backwards at a junction, which was extremely rare in practice). For each model, we evaluate the likelihood under that model of all the binary decisions participants made. For the rewarded-choice model, the learning coefficient, $\alpha$, was chosen such that it maximized the likelihood of responses during the test phase per participant per condition. Mean parameter values across all participants for the Fixed condition were $\alpha = 0.87$, for one-wormhole $\alpha = 0.70$ and, for three-wormholes, $\alpha = 0.73$. We repeated the same exercise for the shortest-distance model. Parameter $\beta$ (Weber fraction) was also fitted per participant per condition. Mean parameter values over participants were: $\beta = 0.23$ for the Fixed condition, $\beta = 0.22$ for one-wormhole and $\beta = 0.33$ for the three-wormhole condition.

Fig 10A shows the two models compared using the data for all participants taken together. In the 'Fixed' condition, the shortest-distance model provides a better account of the data than the rewarded-choice model (negative log likelihood of the shortest-distance model is 60 lower, equivalent to a Bayes Factor of $10^{26}$) whereas, for the three-wormhole condition, the reverse is true and the rewarded-choice model provides a better account than the shortest-distance model (negative log likelihood of rewarded-choice model is 82 lower, equivalent to a Bayes Factor of $10^{-36}$). This change arises because the shortest-distance model becomes a progressively worse predictor of performance for more complex scenes (i.e. from the Fixed to one-wormhole to three-wormhole condition) while the likelihood of the rewarded-choice model changes much less across conditions. An ANOVA on likelihoods per condition per participant confirms that, for the shortest-distance model, condition has a significant effect ($F(2,41) = 14.6$, $p < 0.001$), whereas for the rewarded-choice model there is no significant effect of condition ($F(2,41) = 0.67$, $p = 0.52$). For the shortest-distance model, breaking this main effect of condition down into steps, there is a significant effect of changing from 'Fixed' to one-wormhole condition ($F(1,27) = 5.44$, $p = 0.036$) and from one-wormhole to three-wormhole conditions ($F(1,27) = 9.81$, $p = 0.008$).

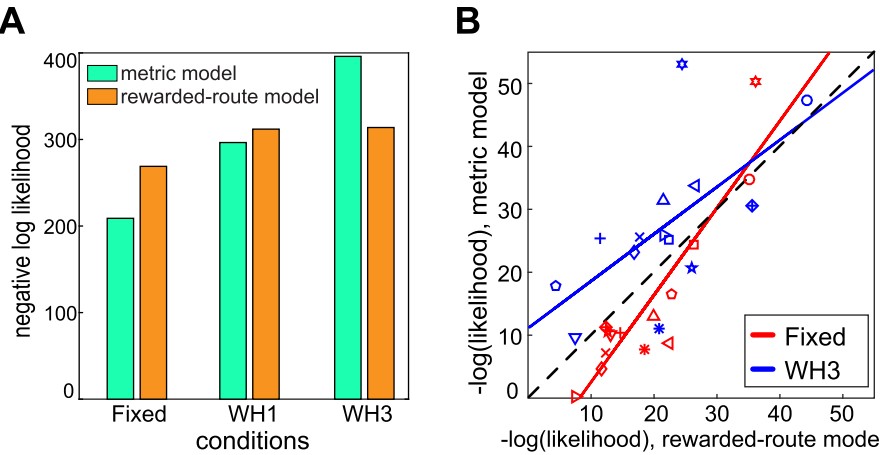

**Fig 10. Model comparison.** A) Likelihoods of combined participants' data per condition per model. B) Comparison of the goodness-of-fit of the two models for each participant in the 'Fixed' and three-wormhole conditions. Different symbols indicate different participants. The same parameters were used for all participants (in both A and B).

Fig 10B illustrates the effect of condition for the shortest route model, shown here for the 'Fixed' and three-wormhole conditions. The negative log likelihood of the data for each participant under the shortest-distance model (plotted on the ordinate) is systematically greater in the three-wormhole (blue) condition: for all but one participant (down-going triangles), the negative log likelihood of the shortest-distance model is greater for the three-wormhole condition than it is for the fixed condition (i.e. for all other pairs in this plot, the blue symbol is higher than the red symbol, paired t-test, t(13) = 5.4, p<0.001).

Another way to assess the significance of the difference in negative log likelihoods between the two models is to sample from each model and then to measure the likelihood of these samples under both models. Fig 11 illustrates why this is an informative way of assessing data under two rival models. Essentially, this is illustrating the fact that data can have a quite high likelihood under two quite different models even when the models are different. In Fig 11, looking only at likelihood of a data point under the orange model (a Gaussian), it can be hard to tell whether a sample was drawn from the orange model (which one would think should provide highly likely samples) or drawn from the green model (which is a quite different Gaussian but gives rise to samples that have a high likelihood under the orange model). By measuring the likelihood of samples drawn from each model and tested against each model, as shown in Fig 11, it is possible to distinguish clearly between the models (following Gootjes-Dreesbach et al, [44]). A likelihood ratio of 1 corresponds to the line of unity on this plot and a data point falling either side of this line favours one model or the other. But this method provides a visualization of whether the data are a typical sample of either model.

Fig 12 shows this type of analysis applied to the shortest-distance and rewarded-choice models. To generate samples from the model, we have used the same number of decision points as there are in the experimental data. At each junction where a participant made a choice in the experiment, a discrete choice was generated from the model according to the probability of a L/R decision in that model for that junction. Hence, a different set of choices is generated for each simulated trial. These samples can then be assessed under each model in the same way as the data. Fig 12 plots the negative log likelihood of each sample under both models for the Fixed, one-wormhole and three-wormhole conditions. Samples drawn from the shortest-distance model are shown in green and from the rewarded-choice model in orange. Unlike Fig 11, the samples from the models in this case are much more likely under

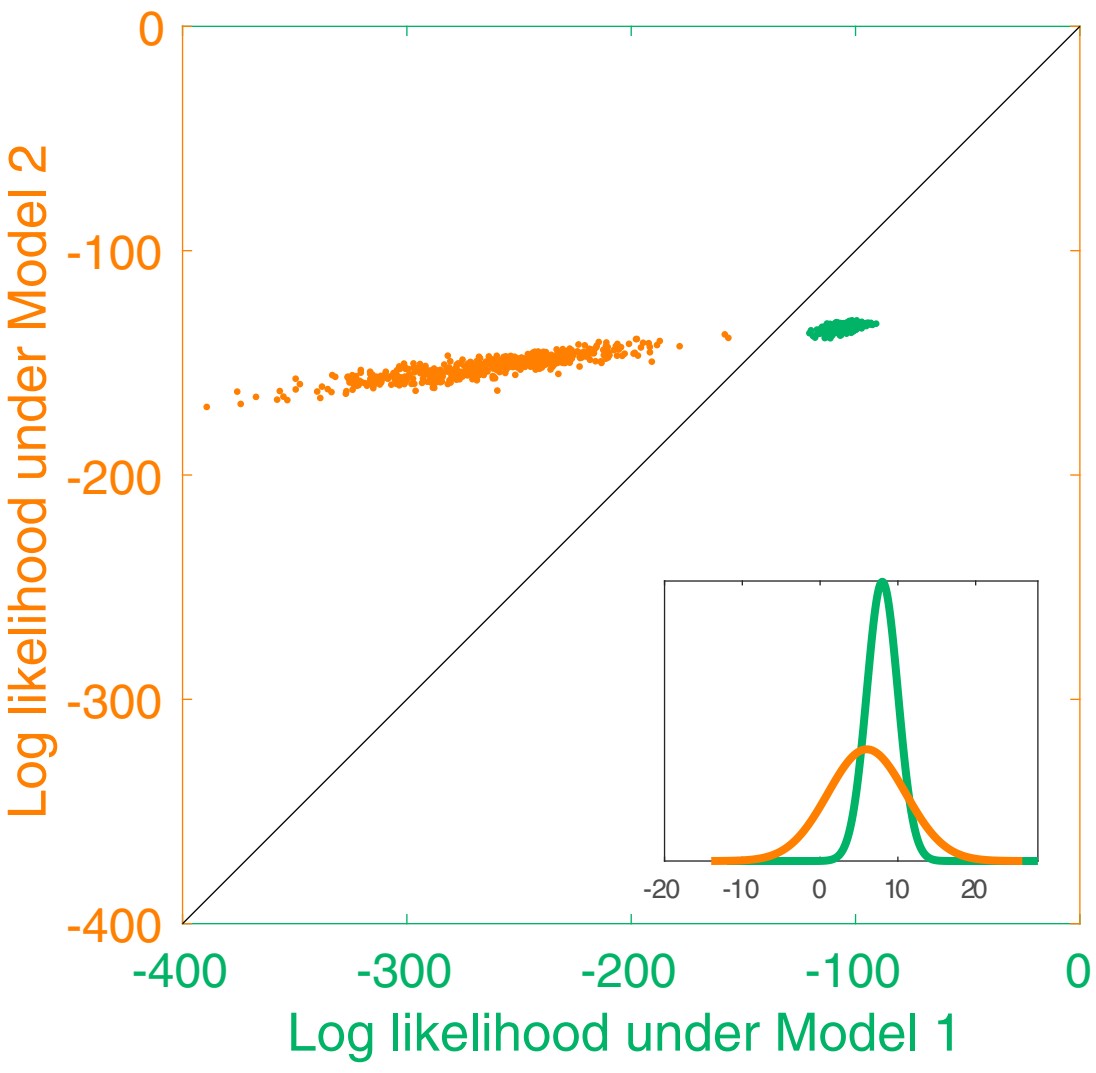

**Fig 11. Illustration of model comparison.** The inset shows two Gaussians as two 'toy' models. Each point in this plot shows the likelihood of 50 samples taken from one model and evaluated under each of the models. Samples taken from either model (Model 1, green Gaussian and green dots, or Model 2, orange Gaussian and orange dots) have similar distributions of likelihoods when evaluated under Model 2 (y-axis values overlap). The reverse is not the case: evaluating the likelihood of the same samples under Model 1 gives rise to quite distinct distributions of likelihoods (x-axis values are distinct).

the model from which they were drawn, suggesting that the models do not overlap in the way that they do in Fig 11. The likelihood of the experimental data (all participants, all Layouts, all repetitions combined) under both models is shown by the grey dot. This data point falls on opposite sides of the line of unity for the Fixed and three-wormhole conditions, which is simply a re-plot of the data from Fig 10A and reiterates the result that the shortest-distance model gives a better account of the data for the Fixed condition while the rewarded-choice model gives a better account for the three-wormhole condition. Data for individual participants is included in the plot (appropriately scaled, see Fig 12 legend), re-plotted from Fig 10B.

It is striking that the likelihood of the combined data (grey dot) is similar to the likelihood of samples taken from the *either* model (i.e. the likelihood of the data falls within the marginal distributions for both models) yet this is not true at all for the samples taken from the models

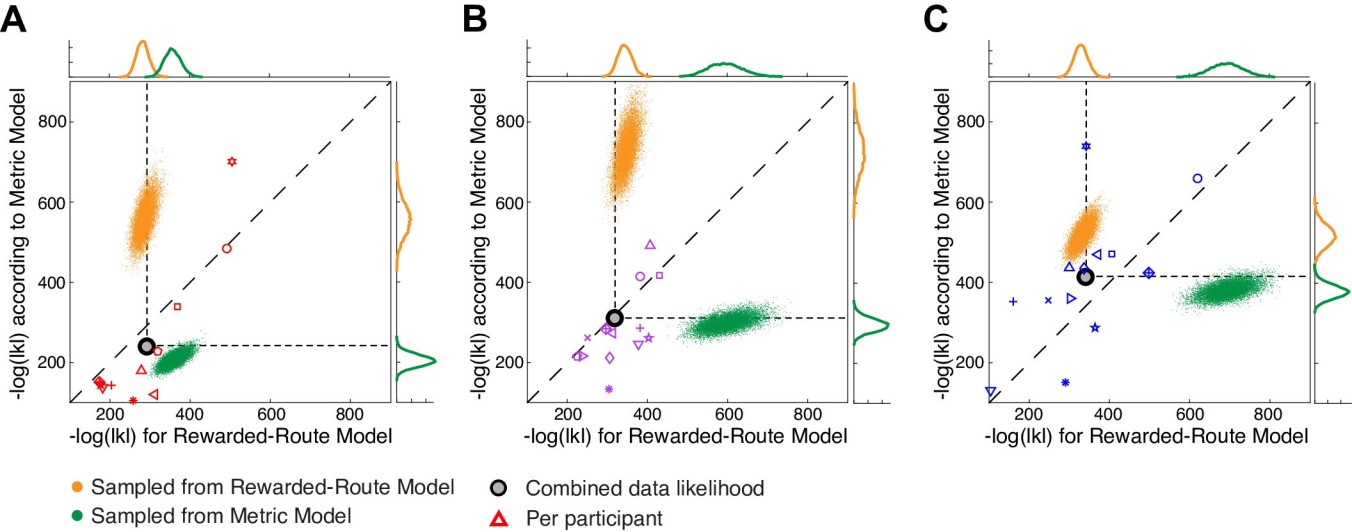

**Fig 12. Comparing models by sampling.** Panels A, B and C show data and models for the Fixed, one-wormhole and three-wormhole conditions respectively. As in Fig 10B, negative log likelihood under the shortest-distance model is plotted against negative log likelihood under the rewarded-choice model. The grey dot in each panel shows, for each condition, the likelihood of the data for all participants in both Layouts and all repeats under both models (with α and β parameters fitted individually per participant). Orange and green clouds indicate 10000 synthetic data-samples generated from the rewarded-choice and shortest-distance models respectively. (likelihoods sampled from individual participant models and then combined). The negative log likelihood of the actual combined data is indicated by the grey circle and is the same as shown in Fig 10A. Open symbols indicate individual participants, where corresponding likelihoods were scaled by raising each to the power n = 14 (number of participants.

(green or orange dots) which are quite likely under the model they were picked from but highly *un*likely under the opposing model. This is because samples from each model include a small number of predictions of decisions that are highly unlikely according to the opposing model. Participants, on the other hand, largely avoid these cases.

We also sampled from a chance model, i.e. where a model participant would choose options at any junction with equal probability. However, this is a highly unlikely model. The chance model gave rise to negative log likelihoods over 2000 for each condition, way outside the range both of participants' data and of our two models.

## 4 Discussion

We have measured the ability of participants to find the shortest route to a previously-viewed target in a virtual labyrinth, especially in cases where the labyrinth has a non-Euclidean structure. Participants' success in this task contrasted markedly with the drastic failures in pointing to previously-viewed targets that we have described before [18] despite the fact that both measures were obtained contemporaneously from the same participants in the same experimental setup. Our main finding is that participants' choices at junctions in the complex, non-physically-realisable, 'wormhole' conditions were predicted by a rewarded-choice model better than a shortest-distance model. In other words, in these wormhole environments, participants tended to make the same choices at junctions that had been successful before when searching for the same target. By contrast, in the simpler, physically-realisable environments participants' choices at junctions were best predicted by a shortest-distance model. Marchette and colleagues [34] described these as 'response' and 'place' strategies respectively. They found that participants spanned a wide range between the two extremes. We found that the relative dominance of the two different strategies changed depending on the complexity of the scene. Within participant, and tested over the same number of trials, we have found evidence that

participants use different strategies or representations depending on the complexity of the scene. Hence, the variation in strategy cannot be due only to individual differences or the number of times an observer experiences an environment [14, 24–27]. Instead, the length of corridors and the number of twists and turns down each seems to have an important effect on the way people tackle the navigation task. This might also be true in a complex environment with many twists and turns that is Euclidean or 'Fixed', without wormholes.

If observers use a graph-like representation, then this change in strategy with different degrees of complexity of the environment is easy to explain. Similar to Siegel and White [22] and others [13], our working hypothesis is that observers start with a representation of connectivity and gradually add information about the edges between nodes. This is a flexible notion. The information about edges could be quite crude (e.g. 'shorter than average edge' versus 'longer distance') but in theory it could include much more precise information. As we discussed in the Introduction, this could include sufficient information about the distance and angles between nodes of the graph representation for it to become impossible to distinguish the behaviour of an observer who relied on this 'well-calibrated' graph from a participant using a Euclidean map, if their tasks were to find shortcuts between (and point between) previously-viewed targets. A similar argument has been made about the representation of object shape [30, 31]. The two types of information that we have explored in this paper, i.e. rewarded choice and shortest distance, can both be seen as part of this hierarchical progression. It makes sense that past success at a junction should be more basic and ranked lower in the hierarchy than distance along an edge (i.e. the latter is part of a more calibrated representation). Our results are compatible with that view: in the more complex, non-Euclidean mazes with longer corridors, observers seem to rely more on previously reward-choices at junctions whereas in the simpler, Euclidean mazes with short corridors observers show evidence that they take account of the lengths of corridors in their choices.

Complexity and non-Euclidean structure co-varied in our experiment because the length of corridors in the maze, and the number of twists and turns (but not junctions) was greater in the non-Euclidean environment. It would take a much larger virtual environment than was available in our lab to disentangle these two. It is worth noting that the likelihoods of the metric model across participants were significantly worse for the three-wormhole condition than for the one-wormhole condition, suggesting that one cannot lump together both the wormhole environments and explain performance simply according to whether an environment has *any* non-Euclidean structure. The three-wormhole condition was more complex and more parts of it were non-Euclidean than the one-wormhole condition, so it is not surprising that the effects of the wormholes were more extreme.

An alternative model, which we have not tested, is that participants take the shortest topological route to the goal (Chrastil and Warren [13]). The fact that wormholes do not affect the topological structure of the maze but radically alter the metric length of certain edges makes this quite a distinct hypothesis from the shortest metric route hypothesis. For example, the shortest topological distance and metric distance between any pair of nodes might correlate highly in the 'Fixed' condition but, assuming this to be the case, the correlation would inevitably be reduced by increasing the metric length of some edges and not others, as happens in the wormhole conditions. Anecdotally, participants in the three-wormhole condition often tend to get lost and have 'loops' in their trajectories in which they return to the same node *en route* to a target. Despite the similar topological structure, this behaviour is uncommon in the 'Fixed' condition.

A speculation that goes beyond our data, but which is testable, is that the same result would be observable in 'fixed' environments of different degrees of complexity even without introducing non-Euclidean elements in the maze such as wormholes. If it were possible to let

participants explore far more complex (but 'fixed', Euclidean) environments and, on other trials, wormhole environments then participants could carry out two tasks simultaneously: (i) search for targets, as in the current experiment, and (ii) judge, in a forced-choice paradigm, whether they believed they were in a complex 'fixed' environment or a 'wormhole' environment. Our prediction is that in a highly complex environment, just like a tourist arriving in a new city, participants would find the second of these tasks quite difficult. We also predict that the rewarded-choice model would be the best model of their navigation strategy for both types of environment during the period of learning when they are unable to discriminate between 'Fixed' and non-Euclidean environments. Once participants have more experience with the environment, they should be able to store information about distances between nodes in their labeled graph. If so, this should enable them to make judgements about the shortest distance between two locations so, at this stage, the shortest-distance model should be the better model for predicting their navigation behaviour. Note that this prediction does not depend on whether or not the participants are able to determine whether the maze is 'Fixed' or non-Euclidean. Such an experiment would establish whether the Euclidean structure of the environment (and, by extension, a Euclidean representation) was important *per se* in determining performance, independent of complexity and familiarity.

Finally, it is worth comparing the navigation data in the current paper to the pointing data in our previous paper collected in the same environment [18], because, unlike the navigation task, pointing is a direct way of testing whether participants can form a Euclidean representation of the scene. Muryy and Glennerster [18] applied different models to the pointing data and concluded that a Euclidean representation could not account for the pointing responses of participants in the three-wormhole condition as successfully as a non-Euclidean one. The non-Euclidean model in that case allowed both the perceived location *and orientation* of the observer to vary as they moved around the maze (yellow bars in Fig 13). The conclusion reached was similar to that in the current paper, i.e. that in the three-wormhole environment participants use a cruder form of representation. In more familiar environments (the 'fixed' condition), participants add information to this representation so that, at its most extreme, the information about each edge in the graph is so rich that the representation is equivalent to full Euclidean structure.

It is logically possible for observers to show excellent performance on the navigation task while making large errors in the pointing task provided one assumes that there is no common, Euclidean representation supporting both tasks. If the visual system relied on a common representation for both tasks, there should be a correlation between the two measures of performance. In each case, we can take measures that indicate how 'lost' a participant is, one from their navigation and one from their pointing. For navigation, we take a ratio of travelled distance to the shortest distance for a full round (including all 4 targets). For participants who are very familiar with the environment, this ratio should be close to one. For pointing, we take the mean absolute pointing error measured for 8 pointing directions (4 targets) at the end of a round as a different measure of how lost they are. In the 'Fixed' condition, there is a significant positive correlation between these two measures, as one might expect (Pearson correlation 0.43, $p < 10^{-9}$). On the other hand, for both wormhole conditions there is no significant correlation (0.02, $p = 0.70$ and 0.07, $p = 0.35$ for WH1 and WH3 respectively), see S4 Fig. This supports the contention that the two measures of 'being lost' are not necessarily linked, something that is compatible with a graph-like representation, but one would not expect this if the observer relied on a Euclidean map for both tasks. There are many examples of such task-dependency in tests of spatial performance: [8, 31, 45–47]. A recent example is the demonstration by Strickrodt et al [20] that participants can point in quite different directions to the same target depending on how they imagine arriving at it [29]. The authors conclude that local

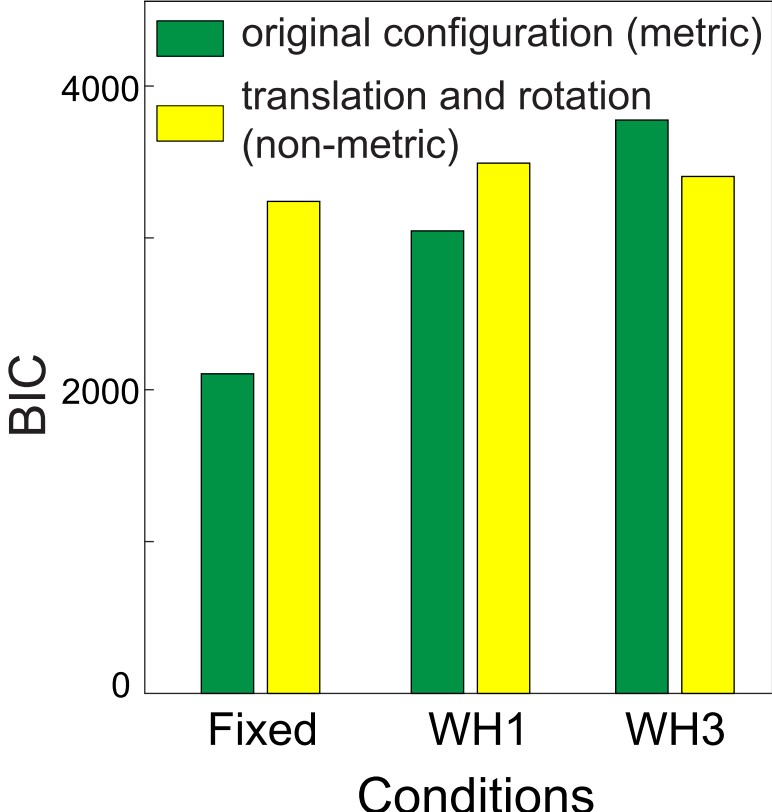

**Fig 13. Data replotted from Muryy and Glennerster [18].** Bayesian Information Criterion is used to compare performance of a metric and a non-metric model of pointing data in the same environment as the current experiment (plotted using data shown originally in Fig 9B in [18]). Unlike the models compared in the current paper, the two pointing models were nested with different numbers of parameters and hence BIC is an appropriate method of comparison.

spatial information is not integrated into a coherent global map. The data we have presented here, especially when considered in conjunction with the pointing data from [18], support this view.

## Supporting information

**S1 Fig.** Schematics of the labyrinths for Layout 1 (A, B, C) and Layout 2 (D, E, F). A) 'Fixed' condition. B) One-wormhole condition; green target is inside wormhole W1. C) Three-wormhole condition; red, green and yellow targets are inside wormholes. The general layout (containing Start, which is marked as 'S', described as the 'ground floor' in the text) remained constant between conditions. The wormholes are marked with letters W surrounded by red and blue triggers. As the participant crossed a trigger, the environment changed without the participant being able to detect this transition, leading to the changes shown in the sub-schematics. Inside a wormhole, the participant could only walk along the route marked by the black dashed line. There were no junctions inside wormholes. D), E), F) show the same for Layout 2. Also see movies for A), B) and C).
(PDF)

**S2 Fig. Travelled distance per round.** Bars show mean distances (in metres) travelled by all participants (n = 14) in each condition. Error bars indicate standard deviations. Horizontal

black lines indicate lengths of the shortest solution, measured along the middle of the corridors. During the 5 rounds of the learning phase, the task was always the same (go from Start to Red-Green-Blue-Yellow). During the test phase (last 3 rounds), participants were asked to solve novel tasks, i.e. the routes were different on every test round.
(PDF)

**S3 Fig. Sketches drawn by participants right after experimental session.** The ground-truth schematics for both scenes in all conditions are shown in S1 Fig.
(PDF)

**S4 Fig. Ability to point accurately against ability to find the shortest path.** The x-axis shows a measure of the ability of participants to find shorter paths: it is a ratio of travelled distance during a full round to the shortest distance of that round. The y-axis shows the ability of participants to point accurately (from Muryy and Glennerster (2018)): this is a mean pointing error (degrees) per round (mean over 8 pointings, since at the end of a round participants pointed 8 times). Solid lines show fitted linear regression models.
(PDF)

**S1 Video. Movie for Fig 1 (first person view of labyrinth, fixed condition).**
(MP4)

**S2 Video. Movie for Fig 2A (trajectory of a participant in the fixed condition).**
(MP4)

**S3 Video. Movie for Fig 2B (trajectory of a participant in the one-wormhole condition).** The structure changes as the participant enters the wormhole.
(MP4)

**S4 Video. Movie for Fig 2C (trajectory for the 3-wormhole condition).**
(MP4)

**S1 Data. Zip of raw data and Matlab.** Data and code to reproduce S2 Fig, ie distance travelled in all conditions by all participants.
(ZIP)

## Author Contributions

**Conceptualization:** Alexander Muryy, Andrew Glennerster.

**Data curation:** Alexander Muryy.

**Formal analysis:** Alexander Muryy.

**Funding acquisition:** Andrew Glennerster.

**Investigation:** Alexander Muryy.

**Methodology:** Alexander Muryy, Andrew Glennerster.

**Project administration:** Andrew Glennerster.

**Resources:** Andrew Glennerster.

**Software:** Alexander Muryy.

**Supervision:** Andrew Glennerster.

**Validation:** Alexander Muryy.

Writing – original draft: Alexander Muryy, Andrew Glennerster.

Writing – review & editing: Alexander Muryy, Andrew Glennerster.

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
