## [Decision Letter · Decision Letter 0]

27 Oct 2020

PONE-D-20-24724

Route selection in non-Euclidean virtual environments

PLOS ONE

Dear Prof Glennerster,

Thank you for submitting your manuscript to PLOS ONE. It has been reviewed by two experts in this field, and they are both very positive about the report. There are, however, a number of minor points that require addressing before it meets the publication criteria and we, therefore, invite you to submit a revised version of the manuscript that addresses their comments. You will see that they are all fair and constructive, and the revisions they suggest will certainly facilitate the paper making an appropriate impact. As such, I encourage you to treat each of them with all due attention.

We look forward to receiving your revised manuscript.

Kind regards,

Alastair D. Smith

Academic Editor

PLOS ONE

2. We noted in your submission details that a portion of your manuscript may have been presented or published elsewhere.

"Figure 11 shows data from Muryy and Glennerster (2018)

Muryy A., Glennerster A. (2018) Pointing Errors in Non-metric Virtual Environments. In: Creem-Regehr S., Schöning J., Klippel A. (eds) Spatial Cognition XI. Spatial Cognition 2018. Lecture Notes in Computer Science, vol 11034. Springer, Cham. https://doi.org/10.1007/978-3-319-96385-3_4

The data are included for comparison with the current data. It is a re-plot of the data, not a copy of the figure."

Please clarify whether this publication was peer-reviewed and formally published. If this work was previously peer-reviewed and published, in the cover letter please provide the reason that this work does not constitute dual publication and should be included in the current manuscript.

3. Your abstract cannot contain citations. Please only include citations in the body text of the manuscript, and ensure that they remain in ascending numerical order on first mention.

Reviewers' comments:

Reviewer's Responses to Questions

**Comments to the Author**

1. Is the manuscript technically sound, and do the data support the conclusions?

Reviewer #1: Yes

Reviewer #2: Partly

2. Has the statistical analysis been performed appropriately and rigorously? 

Reviewer #1: Yes

Reviewer #2: No

3. Have the authors made all data underlying the findings in their manuscript fully available?

Reviewer #1: No

Reviewer #2: Yes

4. Is the manuscript presented in an intelligible fashion and written in standard English?

Reviewer #1: Yes

Reviewer #2: Yes

5. Review Comments to the Author

Reviewer #1: This manuscript reports on wormhole experiments in virtual reality to test predictions of the cognitive map hypothesis compared to the labelled graph hypothesis. The authors interpret the results of several modeling approaches as providing evidence in support of the labelled graph hypothesis. In short, the paths participants took through the environment during learning were predictive of the paths they took at test; and more predictive than the shortest Euclidean path in the wormhole environments.

Full disclosure: I reviewed a previous version of this manuscript for a different journal.

I began to review this paper de novo, but found that its methods and results were largely the same. However, the framing of the paper has changed in a way that addresses nearly all of my original critiques. (Most prominently for me, the authors have shifted the framing away from an emphasis on Euclidean vs. non-Euclidean spaces and toward route complexity).

In sum, I think this is an excellent paper whose data are aligned with its conclusions. It is complex, but that’s not a criticism. Readers who spend the time to understand what is going on will be rewarded.

My only (relatively small) critique is that the figures and data are quite complex. The authors do an admirable job breaking things down, but certain figures and their descriptions are difficult. For example, Figure 10 might be better broken down into multiple parts within each panel (a/b/c). If I’m understanding correctly, the authors are showing that on each sample the explanatory model that sample was drawn from was more likely to be the model it was drawn from? Seems logical enough, but I’m having a hard time tying in how the participant’s data maps onto that. It could be I’m missing something from the text, but these figures are quite busy and Fig. 10 in particular could be explained more clearly.

I think readers would also benefit from a figure or video illustrating how the two models operate. The shortest-distance model is clear; but what exactly the “rewarded-route” model is doing and even why it’s called the rewarded-route become easily lost in a paper that’s loaded with terms, abbreviations, etc.

Two other minor points:

1. I did not see a discussion of sample size. How was 14 participants decided? I also see a high proportion of participants were excluded. Do the authors have any data on these subjects that might show them to be different (e.g., self-report, gender, etc.? )

2. I did not find the data available for this study anywhere in the manuscript or supplementary materials.

Reviewer #2: This is an interesting manuscript that investigates the knowledge basis for route selection by cleverly manipulating the complexity of non-Euclidean environments. I like this line of research. I particularly appreciated the analysis in the Discussion showing that route choices and metric pointing responses do not correlate in these environments. However, I do have some questions, both theoretical and methodological, about the results and their interpretation. I think the following comments need to be addressed before the paper can be accepted for publication.

1. The authors argue that the shift away from metric path to rewarded route depends on environmental complexity, but unfortunately complexity is confounded with Euclidean and non-Euclidean environments. It is thus critical to compare the goodness-of-fit for the two models between WH1 and WH2 (Figure 9A), which differ only in complexity. As far as I can see, they authors find a main effect of environment for the metric model fit, but do not make statistical comparisons between WH1 and WH2. These should be reported.

2. The authors seem to assume that rewarded-route learning yields knowledge of the “connectivity” of the environment. But they fail to distinguish between route knowledge (place-goal-action associations) and topological graph knowledge (enabling novel detours to the same goal, e.g. Chrastil & Warren, 2014). This difference should be described in the introduction (Lines 95-106) and acknowledged as an alternative in the Discussion (p. 22-23) – see next comment. Does a reinforcement learning algorithm only yield route knowledge, or could it produce topological graph knowledge – true connectivity?

3. The authors report a shift away from the metric path model, and toward the rewarded route model, with environmental complexity. But there is a third hypothesis: the shortest topological path (number of edges), which depends on connectivity but not edge weights. Can the present data distinguish a topological path model from the other two? What is the correlation between topological distance and metric distance in their mazes, and between topological distance and rewarded route?

4. In the Introduction (Lines 91-92) the authors say that a labeled graph with precise and internally consistent local metric information “is indistinguishable in practice from a Euclidean map.” I don’t think this is correct. The difference is that the local information is not embedded in a global coordinate system, so the graph has nothing to say about the spatial regions in between its nodes and edges. Clever experimental tasks might be designed to probe knowledge of those regions.

5. Lines 159-160: I’m concerned that eyeheight was only 1m above the ground plane. I understand that the entire environment scaled proportionally, but if actual eyeheight can be perceived (e.g. from binocular cues) then visual and idiothetic information for metric distances would be inconsistent, which may interfere with learning metric path lengths.

Details

• Line 16, Abstract: Is Muryy & Glennerster (2018) published?

• Lines 135-6: “We called physically possible mazes ‘Fixed’, for short.” Did you use this term with the subjects, or are you just using it in the manuscript for the reader’s benefit? That is, were the subjects aware of which maze was ‘Fixed’ and which not?

• Line 153: What was the motion-to-photon latency of the HMD?

• Lines 265-266: “For different repetitions of the same Layout, the structure of the labyrinth and target locations were identical, but the colours of the targets were changed.” If the target colors changed on every repetition, how could subjects ever learn the connectivity of the environment? Or were the target colors always the same in each Environment (Fixed, WH1, WH2)?

• Lines 537-542: These two sentences seem inconsistent. The authors predict both that the rewarded-route model, and that the shortest-distance model, would be the best model when subjects are unable to judge between ‘Fixed’ and non-Euclidean environments.

• On Lines 84-85 the authors say that graph structure and local metric information can be acquired in parallel, contrary to Siegel & White (1975), but consistent with Ishikawa & Montello (2006) and Warren (2019). Then on Lines 519-520 they say that observers start with connectivity and then add local metric information, consistent with Siegel & White (1975). This seems inconsistent, and their data do not support one or the other: some metric information could be acquired even in WH2.

6. PLOS authors have the option to publish the peer review history of their article (what does this mean?). If published, this will include your full peer review and any attached files.

Reviewer #1: **Yes: **Steven M. Weisberg

Reviewer #2: No

---

## [Author Response · Author response to Decision Letter 0]

21 Dec 2020

These replies are also in the attached file with response to reviewers.

Done

and

Done

2. We noted in your submission details that a portion of your manuscript may have been presented or published elsewhere.

"Figure 11 shows data from Muryy and Glennerster (2018)

Muryy A., Glennerster A. (2018) Pointing Errors in Non-metric Virtual Environments. In: Creem-Regehr S., Schöning J., Klippel A. (eds) Spatial Cognition XI. Spatial Cognition 2018. Lecture Notes in Computer Science, vol 11034. Springer, Cham. https://doi.org/10.1007/978-3-319-96385-3_4

The data are included for comparison with the current data. It is a re-plot of the data, not a copy of the figure."

Please clarify whether this publication was peer-reviewed and formally published. If this work was previously peer-reviewed and published, in the cover letter please provide the reason that this work does not constitute dual publication and should be included in the current manuscript.

 Yes, the Lecture Notes in Computer Science paper, Muryy and Glennerster (2018), is a peer-reviewed publication. 

The previous Fig 11 was not reproduced from Muryy and Glennerster (2018) but it did re-plot some of the data shown in that paper in a way that makes it easy to compare with the data from the current paper. It is an important comparison we are making here (between the performance of participants on two different tasks in the same experiment). It is not dual publication because we do not claim that the data are newly published here, we are merely drawing the reader’s attention to the similarity between an aspect of the data in Muryy and Glennerster (2018) and the data in the current paper. This requires re-plotting the data to draw the link between the two and it helps the reader to have the two sets of data side by side to make the comparison.

3. Your abstract cannot contain citations. Please only include citations in the body text of the manuscript, and ensure that they remain in ascending numerical order on first mention.

Citation removed from the abstract.

---

## [Decision Letter · Decision Letter 1]

16 Feb 2021

Route selection in non-Euclidean virtual environments

PONE-D-20-24724R1

Dear Prof Glennerster,

Thank you for submitting the revised version of your manuscript. It has been reviewed by the same experts as before and I am pleased to say that it has been judged scientifically suitable for publication. It will, therefore, be formally accepted for publication once it meets all outstanding technical requirements.

With kind regards,

Alastair D. Smith

Academic Editor

PLOS ONE

Reviewers' comments:

Reviewer's Responses to Questions

**Comments to the Author**

1. If the authors have adequately addressed your comments raised in a previous round of review and you feel that this manuscript is now acceptable for publication, you may indicate that here to bypass the “Comments to the Author” section, enter your conflict of interest statement in the “Confidential to Editor” section, and submit your "Accept" recommendation.

Reviewer #1: All comments have been addressed

Reviewer #2: All comments have been addressed

2. Is the manuscript technically sound, and do the data support the conclusions?

Reviewer #1: Yes

Reviewer #2: Yes

3. Has the statistical analysis been performed appropriately and rigorously? 

Reviewer #1: Yes

Reviewer #2: Yes

4. Have the authors made all data underlying the findings in their manuscript fully available?

Reviewer #1: Yes

Reviewer #2: Yes

5. Is the manuscript presented in an intelligible fashion and written in standard English?

Reviewer #1: Yes

Reviewer #2: Yes

6. Review Comments to the Author

Reviewer #1: Excellent work. Congratulations. The manuscript meets all requirements. [I am obligated to fill out a character count of 100 for this response box. That is the rest of this text.]

Reviewer #2: (No Response)

7. PLOS authors have the option to publish the peer review history of their article (what does this mean?). If published, this will include your full peer review and any attached files.

Reviewer #1: **Yes: **Steven M Weisberg

Reviewer #2: No

---

## [Editor Report · Acceptance letter]

7 Apr 2021

PONE-D-20-24724R1 

Route selection in non-Euclidean virtual environments 

Dear Dr. Glennerster:

I'm pleased to inform you that your manuscript has been deemed suitable for publication in PLOS ONE. Congratulations! Your manuscript is now with our production department. 

Kind regards, 

on behalf of

Dr Alastair Smith 

Academic Editor

PLOS ONE